# Phylogenetic Relationships of Turkish Indigenous Donkey Populations Determined by Mitochondrial DNA D-loop Region

**DOI:** 10.3390/ani10111970

**Published:** 2020-10-27

**Authors:** Emel Özkan Ünal, Fulya Özdil, Selçuk Kaplan, Eser Kemal Gürcan, Serdar Genç, Sezen Arat, Mehmet İhsan Soysal

**Affiliations:** 1Faculty of Agriculture, Department of Animal Science, Tekirdağ Namık Kemal University, Tekirdağ 59030, Turkey; egurcan@nku.edu.tr (E.K.G.); misoysal@nku.edu.tr (M.İ.S.); 2Faculty of Agriculture, Department of Agricultural Biotechnology, Tekirdağ Namık Kemal University, Tekirdağ 59030, Turkey; sarat@nku.edu.tr; 3Faculty of Veterinary Medicine, Tekirdağ Namık Kemal University, Tekirdağ 59030, Turkey; skaplan@nku.edu.tr; 4Faculty of Agriculture, Department of Agricultural Biotechnology, Kırşehir Ahi Evran University, Kırşehir 40100, Turkey; serdar.genc@ahievran.edu.tr

**Keywords:** Turkish donkeys, mtDNA, D-loop, genetic diversity, haplotype, maternal origin

## Abstract

**Simple Summary:**

This paper represents the first fundamental report of mtDNA diversity in Turkish indigenous donkey breeds and presents findings for the origin and genetic characterization of donkey populations dispersed in seven geographical regions in Turkey, and thus reveals insights into their genetic history. The median-joining network and phylogenetic tree exhibit two different maternal lineages of the 16 Turkish indigenous donkey populations.

**Abstract:**

In this study, to analyze the mtDNA D-loop region and the origin of the maternal lineages of 16 different donkey populations, and to assess the domestication of Turkish indigenous donkeys in seven geographical regions, we investigated the DNA sequences of the D-loop region of 315 indigenous donkeys from Turkey. A total of 54 haplotypes, resulting from 35 polymorphic regions (27 parsimoniously informative and 6 singleton sites), were defined. Twenty-eight of these haplotypes are unique (51.85%), and 26 are shared among different Turkish indigenous donkey populations. The most frequent haplotype was Hap 1 (45.71%), followed by two haplotypes (Hap 4, 15.55% and Hap 7, 5.39%). The breed genetic diversity, evaluated by the haplotype diversity (H_D_) and nucleotide diversity (π_D_), for the Turkish donkey populations ranged from 0.533 ± 0.180 (Tekirdağ–Malkara, MAL) to 0.933 ± 0.122 (Aydin, AYD), and from 0.01196 ± 0.0026 (Antalya, ANT) to 0.02101 ± 0.0041 (Aydin, AYD), respectively. We observed moderate-to-high levels of haplotype diversity and moderate nucleotide diversity, indicating plentiful genetic diversity in all of the Turkish indigenous donkey populations. Phylogenetic analysis (NJT) and median-joining network analysis established that all haplotypes were distinctly grouped into two major haplogroups. The results of AMOVA analyses, based on geographic structuring of Turkish native donkey populations, highlighted that the majority of the observed variance is due to differences among samples within populations. The observed differences between groups were found to be statistically significant. Comparison among Turkish indigenous donkey mtDNA D-loop regions and haplotypes, and different countries’ donkey breeds and wild asses, identified two clades and which is named Somali (Clade IV) and Nubian (Clade V) lineages. The results can be used to understand the origin of Turkish donkey populations clearly, and to resolve the phylogenetic relationship among all of the different regions.

## 1. Introduction

Animal domestication performed a substantial role in the occurrence of modern life and urbaneness. The definition of genetic origin and phylogenetic relations between animals is crucial for comprehension of cultural alterations in these civilizations, and eventually has public and scientific desirability [1,2]. Genetic origin is described using three particular genetic resources; mitochondrial genome, Y chromosome, and nuclear genome [3]. Mitochondrial DNA (mtDNA) is frequently involved in species definition and genetic differentiation of domestic animals, due to its particular characteristics, for instance; well-recognized gene conformation, more copies than nuclear DNA, absence of introns, high polymorphism rate, maternal inheritance, and lack of recombination circumstances [4,5,6]. Additionally, frequently it is the sole genetic resource that can be acquired from degraded or archaeological biological fossils. mtDNA has been used to research the change of modern domestic animals, i.e., to define wild ancestors and to identify domestication centers, to search the evolutionary relationships, genomic complexity, genetic diversification, and the origins of several domestic animals [7,8,9]. MtDNA, containing exons, non-coding regions, and introns, spans 15–20 kb in different mammalian species. The non-coding region is the control region (CR), also called the displacement-loop region (D-loop) [10]. It has a higher mutation rate than the cytochrome b (*Cyt-b)* gene. In donkeys, the span of the control region of the mtDNA, except for the repetitive regions, is 959 base pairs long, it is also one less nucleotide than the horse’s mtDNA. Between the donkey and the horse mtDNA there are 108 variations (11.2%), in an alignment of the two sequences, there are 88 transitions, 16 transversions, and four insertion/deletions (indels) [11]. For this reason, largely these regions have been used to analyze the maternal origin and genetic diversity of domestic donkeys [12,13].

The period of domestication and distribution of the donkey (*Equus asinus*) is especially interesting for reason that the donkey has been substantially used by humans to transport materials and people from up to as long as roughly 5000–7000 years ago [14,15]. The domestication centers and the lineages of the donkey are still disputable, but the obviously admitted hypothesis, depending on archaeological, historical, and ethnographical resources, suggests that the domestic donkey essentially has its origin in the Nubian (*Equus africanus africanus*) and Somalian (*E. a. somaliensis*) ass. Two mitochondrial haplogroups lineages, clade 1 and clade 2, were defined in analyses on modern domestic donkeys. Therefore, giving evidence for two unconnected domestication processes in donkeys [16,17,18]. clade 1 is thought to derive from the Nubian wild ass (*E. a. africanus*) [16,18], whereas Kimura et al. [18] and Kefena et al. [19] in recent years proposed that clade 2 donkeys may be descendants of a yet unknown extinct, wild population rather than the Somali wild ass. Beja-Pereira et al. [16] defined high levels of genetic variation in both the lineages in Northeast African territories, and deduced that this region was one of the main centers of donkey domestication. However, Kimura et al. [18] informed that the mtDNA sequences acquired from the extant Somali wild ass were categorized together with the formerly defined Somali wild ass specimens, and failed to indicate any sequence similarity with domestic donkeys of both clades. This study suggested the existence of another ancestor of the domestic donkeys of clade 2, depending on an additional, still unapproved, extinct wild population in the Northeast African territories. In another study, depending on genetic, linguistic, and zooarchaeological evidence it was affirmed that Ethiopia could be one of the key areas of donkey variety and domestication [19].

Donkeys are a former constituent of Turkey’s domestic animal community. They had a crucial role in the country’s agricultural economy for hundreds of years. Generally, as in most of the world in the 21st century, the donkey in Turkey has a depreciated status as a center of the rural economy, and has been supplanted by mechanical means of transport and power [20]. In ancient times, the donkey played an important part in agriculture and transportation, particularly in the rural areas of Turkey. Over the last 30 years, as farm mechanization and traffic automation have become popular, the number of donkeys has reduced strikingly in the world [14].

In many developed countries the advent of “conservation” strategies has resulted in many native breeds of many domestic species being described genetically and morphologically [7,21,22,23]. Parts of Asian, European, and African donkey’s mtDNA variation and genetic diversity have been described [4,12,15,16,18,24,25,26,27,28].

The objective of this study was: (i) to determine the genetic variability between seven different geographical regional donkey populations in Turkey, (ii) to investigate the genetic structure of Turkish native donkey populations and to explore the maternal origin, with mtDNA D-loop sequences, and explore the domestic history of Turkish indigenous donkey populations, (iii) to evaluate genetic breed sub-structuring, and (iv) the comparison of mitochondrial DNA sequences of Turkish populations with other countries’ domestic donkey breeds and wild asses. The results will be used to better understand the Turkish donkey population’s origin, and to resolve the phylogenetic relationship among all the different regions.

## 2. Materials and Methods

### 2.1. Population Determination, Sample Collection, and DNA Isolation

According to the FAO database (Domestic Animal Diversity Information System, accessed 15 September 2019); the estimated number of donkeys in Turkey is about 150,000. Three types of donkey are recognized in Turkey (FAO/DADIS 2012). The Anatolian is found all over the country, not just on the central plateau, and is usually grey or black in color. The Merzifon or Marsovan, from the town and district of the same name in Amasya province in the central Black Sea region, is at risk. The Karakaçan type, presumably named after the Karakaçan nomads of the Balkans and Thrace, is also considered at risk (FAO-DADIS 2012) [20].

In this study a total of 315 blood samples were collected from Turkish indigenous donkeys raised in seven different geographical regions, from 16 different provinces of Turkey (Figure 1). These locations were selected to represent the expected native Turkish donkey breeds: Anatolian, Merzifon, Karakaçan, etc. The provinces and the potential geographical dispersion of some Turkish indigenous breeds, with the number of animals, are given in Table 1. These provinces of Turkey were known for their relatively large donkey populations.

From those donkey samples, 10 mL of whole blood was taken from the jugular vein with EDTA-coated vacutainer tubes. Genomic DNA was isolated by using a standard phenol-chloroform-isoamyl alcohol extraction method [29]. The concentration of DNA was analyzed to compare with the standard DNA marker concentration on agarose gels. The quality and quantity of DNA was checked on 0.6% agarose gels, prepared with a Tris-Boric acid-EDTA buffer. The isolated DNA was diluted with TE buffer (10:1) and stored at +4 °C till polymerase chain reaction (PCR) analysis.

### 2.2. Amplification of mtDNA Control Regions (D-Loop) and Sequencing

The 383 bp segment, from the mitochondrial DNA control region (D-loop), from the donkey genome, was amplified by PCR, as described by Aranguren-Méndez et al. [4]. The primer sequences were amplified between the sites 15,387 to 15,769 bp. The PCR conditions were optimized for all the primer pairs selected for the study. mtDNA D-loop was amplified by PCR, carried out in 20 µL reaction volume, containing 50 ng of genomic DNA, 2.0 mM MgCl_2_, 0.2 mM each of dNTP, 0.5 µM of each primer, 1 X PCR buffer, and 1 U of Taq DNA polymerase. Amplification was performed using a BIORAD MyCycler Thermal Cycler (Bio-Rad Laboratories, Inc, Berkeley, CA, USA) with the following conditions: initial denaturation at 94 °C for 5 min followed by 35 cycles of denaturation at 94 °C for 30 s, annealing at 58 °C for 30 s, extension at 72 °C for 2 min, and final extension at 72 °C for 15 min. The amplified products were electrophoresed, visualized under UV radiations, and purified to be sequenced using a ABI prism 3500 genetic analyzer. PCR products were sequenced for both directions. The raw sequence trace files were checked for the presence of ambiguous bases using ChromasPro v. 1.7.4 software.

### 2.3. Data Analyses

The obtained sequences were controlled for the presence of ambiguous bases by using the software, ChromasPro v.1.7.4. For identifying the haplotypes (Hap), all sequences were aligned with the complete donkey mtDNA sequences, MK896308 [30] and X97337 [11], commonly used as reference sequences, using the ClustalW algorithm implemented in BioEdit v7.0.9 [31] and MEGA v.7.0.26 software [32]. The number of haplotypes (N_H_), number of private haplotypes (P_H_), haplotype diversity (H_D_), total polymorphic sites (S), parsimony informative sites (SPI), singleton site (S_S_), shared haplotype (S_H_), the nucleotide diversity (π**_D_**), and average number of nucleotide differences (k) were estimated using DNASP 6.12.03 ×64 software [33]. Pairwise population genetic differentiation (F_ST_) and the average number of pairwise differences within and between populations were calculated using Arlequin v.3.5.2.2 [34]. In order to define the genetic relationships between Turkish indigenous donkey populations mtDNA sequences and those available in Genbank for other donkey populations originated from European, Asian, and African origin domestic donkeys, as well as Ethiopian donkeys, and wild asses, a total of 429 sequences were considered. All the sequences were trimmed at 344 bp to maximize the number of sequences included. The mtDNA sequences from the GenBank database used for haplogroup definition and relationship investigation are reported in Appendix A. To evaluate the genetic variance among donkeys within and between populations, we performed the analysis of molecular variance (AMOVA) using Arlequin v.3.5.2.2 [34]. MEGA v.7.0.26 software was used to analyze the relationships among the haplotypes identified in Turkish native donkeys [32]. The T-REX web server was used to analyze the relationships among the haplotypes identified in Turkish native donkey populations by geographical regions. Genetic relationships among populations were reconstructed using median-joining networks (MJN) using Network v.10.1.0.0 software [35]. Whole mtDNA D-loop sequences identified in this research were submitted to the GenBank with the accession numbers MH683672-MH683725.

## 3. Results

### 3.1. Genetic Diversity of the mtDNA D-Loop Analyses of Turkish Indigenous Donkeys

The mtDNA D-loop sequences were taken for all analyzed 315 samples from seven different regions, and from 16 different provinces of Turkey. In Table 2, the haplotype and nucleotide diversities of the sampled populations are presented. The sequencing results of mtDNA D-loop of Turkish indigenous donkey populations revealed 54 different haplotypes (Figure 2a), which were designated Hap 1 to Hap 54, (GenBank accession nos. MH683672–MH683725) resulting from 35 polymorphic sites. Of the polymorphic sites, there were 35 transitions and 2 transversions, and two sites had both a transition and transversion, suggesting a strong bias toward transitions. No insertions or deletions were observed in Turkish donkey populations. The distribution of the samples to haplotypes is presented in Appendix A. The haplotype distribution, by clusters corresponding to the Turkish indigenous donkeys, is shown in Figure 2b. Construction of the phylogenetic tree by the maximum likelihood method showed two different clusters. The maximum composite likelihood estimates of the nucleotide variation pattern, determined for the 315 sequences, were 33.00% (A), 27.3% (T), 28.6% (C), and 11.1% (G). Positions containing gaps and missing data were eliminated. Of the 54 haplotypes, 28 are unique and 26 are shared among different Turkish indigenous donkey populations. The most frequent haplotype is Hap 1 (45.71%), with 144 samples from 16 different populations, followed by two haplotypes (Hap 4—15.55% and Hap 7—5.39%), each with more than 10 samples. Hap 1, determined in 45.71% of donkeys (n = 144), had the highest frequency in the studied sample, while in the South East Anatolia (SAR) region donkey population, its frequency was 23.61% (n = 34), in Marmara (MRM), 22.91% (n = 33), in Mediterranean (MDR), 18.75% (n = 27); in Black Sea (BSR), 15.27% (n = 22), in Eastern Anatolia (EAR), 12.5% (n = 18), in Aegean (AER) 5.56% (n = 8), and in Central Anatolia (CAR), 1.38% (n = 2). High frequency (15.55%, n = 49) and representation in seven region populations (SAR, 32.65%, n = 16; AER, 18.36%, n = 9; MRM, and CAR, 12.24%, n = 6; in MDR, 10.20%, n = 5; in EAR, 8.16%, n = 4; in BSR, 6.12%, n = 3) was defined for Hap 4. In total, only three haplotypes were present in all groups; these are the abovementioned Hap 1, Hap 4, and Hap 7.

In Table 3, the observed haplotypes, their annotation, and frequencies in studied populations are presented. The 26 haplotypes were shared between populations (Hap 1, Hap 2, Hap 4–7, Hap 20, Hap 23, Hap 25–27, Hap 29, Hap 32–34, and Hap 37), whereas the remaining 28 were unique. Haplogroup Hap 1 is thought to be the oldest haplogroup. The regions with the highest number of haplotypes were the Southeast Anatolia (21), Marmara (19), Mediterranean (18), and Black Sea (17) regions. Southeast Anatolia populations have higher mtDNA diversity with 21 haplotypes. Eleven of these haplotypes were observed only in Southeast Anatolian populations. It was observed that the lowest haplotype diversity is in Central Anatolia and Eastern Anatolia. It is thought that the reason for this decrease was due to the low number of samples, and the sampling of only one province in these regions. The haplotype number in each population varies from 4 to 15. The highest haplotype number was determined in Mardin (MRD) donkeys followed by Amasya-Merzifon (MER), Kırklareli (KIR), and Antalya (ANT) donkey populations. The lowest haplotype number was determined in (: Tekirdağ–Malkara (MAL), Kütahya (KUT), Kahramanmaraş (KRM)) donkeys followed by Tokat (TOK) and Aydın (AYD) donkey populations.

The breed genetic diversity (Table 2), evaluated by the haplotype diversity (H_D_) and nucleotide diversity (π_D_), for the Turkish indigenous donkey populations ranged from 0.533 ± 0.180 (MAL)–0.933 ± 0.122 (AYD), and 0.01196 ± 0.0026 (ANT)–0.02101 ± 0.0041 (AYD), respectively. The overall values for haplotype and nucleotide diversities were 0.763 ± 0.023 and 0.01656 ± 0.00046, respectively (Table 1). Analysis of mtDNA D-loop region for each population revealed that the AYD donkey population had the highest haplotype diversity followed by the KAS donkey population. In MAL donkeys, the lowest haplotype diversity was obtained, followed by KAR and SAN donkey populations. AYD donkeys had the highest estimates of nucleotide diversity followed by the ISP donkey population. In general, nucleotide diversities ranged from 0.01196 in ANT donkeys to 0.02101 in AYD donkeys. The highest number of private haplotypes (n = 8) was observed in the KRM population. Additionally, the average number of nucleotide variation (k) ranged from 4.172 (ANT) to 7.333 (AYD) with an average of 5.745.

The 54 haplotypes were defined by 29 parsimony informative sites. Parsimony informative sites [32] are defined as mutations that have a minimum of two nucleotides that are present at least twice in the sampled population, whereas noninformative sites are singleton sites. Of the polymorphic sites, there were 35 transitions and two transversions, and two sites had both a transition and transversion, suggesting a strong bias toward transitions. No insertions or deletions were observed. At a population level, all the breeds showed one or more singleton sites. In Table 2, the CAT population counted the highest number of non-informative singleton sites (S_S_), while the lowest number of polymorphic sites (S) was identified in MAL and KRM populations. In addition, the mean number of nucleotide differences (k) was the highest for AYD donkeys (7.333) followed by ISP donkeys (6.689), while ANT donkeys had the lowest mean number of nucleotide differences (4.172). The AYD breed observed five haplotypes, the highest π value (0.02101), the highest number of haplotype diversity (0.933), and the highest nucleotide diversity (k = 7.333). The high k value for the AYD breed is due to the high difference among the five haplotypes identified in the population. Despite the low number of samples, a high H_D_ and (π) value, with five haplotypes (Hap 1, Hap 4, Hap 7, Hap 14, and Hap 25) for the AYD was found (Table 1). However, ANT have the lowest π**_D_** value (0.01196) and k value (k = 4.172), but haplotypes have a high H_D_ value (0.717).

The matrix of pairwise F_ST_ (fixation index) value and D_A_ distances value between Turkish indigenous donkey populations from seven geographical regions are shown in Table 4. The F_ST_ values range from 0.0302 (Aegean vs. Central Anatolia) to 0.2523 (Marmara vs. Central Anatolia). The highest differences in F_ST_ values were found between the Marmara and Central Anatolia region populations (F_ST_: 0.2523, *p* < 0.01), followed by between the Mediterranean and Central Anatolia region donkey populations (F_ST_: 0.2130, *p* < 0.01). In contrast, there is little genetic differentiation between Southeastern Anatolia and Black Sea donkeys (F_ST_:0.0004), followed by between Mediterranean and Black Sea (F_ST_:0.0068). In the study, some F_ST_ values were observed to be negative (Table 4). Negative F_ST_ values should be effectively seen as zero values. A zero value for F_ST_ means that there is no genetic subdivision between the populations considered. In addition, the matrix of pairwise F_ST_ values between Turkish indigenous donkey populations from 16 different provinces is revealed in Figure 3.

The F_ST_ ranged from −0.1184 (TOK-MUG) to 0.3446 (KUT-KRM). Out of 120 pairwise F_ST_ comparisons, 28 had F_ST_ between 0 and 0.05, showing little genetic variation, while 16 comparisons had F_ST_ between 0.05 and 0.15, revealing moderate genetic variation, 13 comparisons had F_ST_ between 0.15 and 0.25, indicating major genetic variation, and eight comparisons had F_ST_ greater than 0.25. Negative F_ST_ were registered in some comparisons and these equate to zero F_ST_ (Appendix A). The highest F_ST_ (>0.25) values were seen between AER, MRM, MDR, BSR, and CAR region populations–KUT-KIR, KUT-CAT, KUT-KRM, KRM-AYD, KRM-KAS, KAS-CAT, KON-CAT, and KON-KRM. These populations showed values corresponding to very great genetic differentiation (Figure 3).

A Neighbor-joining tree (NJT) from Reynolds’ linearized genetic distances (Appendix A) for 315 tested sequences from different regions was constructed using the T-REX web server. The NJT clearly demonstrated three distinct clades in Turkish donkey populations from seven geographical regions (Figure 4). Three geographical regions, including CAR, AER, and SAR, clustered in clade I, while a different three geographical region populations, including MDR, MRM, and EAR, clustered in clade II. Clade III included only the Black Sea region populations (BSR). The topology of this tree reflected the patterns of Reynolds’ linearized pairwise genetic distances, where MRM and MDR, MRM and EAR, BSR and EAR, CAR and AER, MDR and EAR, as well as BSR and SAR region donkeys were genetically much closer to each other than to the rest of the pairwise genetic distances (F_ST_: Appendix A).

The median-joining network of mtDNA D-loop region was described between Turkish donkey populations (Figure 5). The size of the provided node is proportional to the number of samples represented in a haplotype, with the smallest node representing a single individual. Branch length is proportional to the mutational distance; only mutational distances greater than one are indicated. In Figure 5, two major clades were obtained from the reconstructed median joining network. The clade 1 lineage was obtained from 31 haplotypes (Hap 4, Hap 6, Hap 7, Hap 10, Hap 12, Hap 16, Hap 18, Hap 20, Hap 22–27, Hap 29, Hap 31, Hap 33, Hap 34, Hap 40, Hap 41, Hap 43–47, and Hap 49–54). This clade was derived mainly from haplotypes obtained from SAR (n = 32), AER (n = 22), BSR (n = 20), MRM (n = 15), MDR (n = 15), CAR (n = 10), and EAR (n = 9) region donkey populations. The clade 2 lineage was formed from 23 haplotypes (Hap 1–3, Hap 5, Hap 8, Hap 9, Hap 11, Hap 13–15, Hap 17, Hap 19, Hap 21, Hap 28, Hap 30, Hap 32, Hap 35–39, Hap 42, and Hap 48) and it was separated from clade 1 by five mutation steps (Figure 5). This clade was dominated by haplotypes received from MRM (n = 45), MDR (n = 38), SAR (n = 38), BSR (n = 30), EAR (n = 22), AER (n = 13), and CAR (n = 5) region donkey populations. Star-like clusters were seen around the haplotypes Hap 1 and Hap 4 in the Median joining (MJ) tree and localization of the haplotypes (Figure 5). Figure 6 show that Clade 1 and Clade 2 distribution more clearly.

### 3.2. Turkish Donkey Populations Comparing with Other Donkey Populations

Comparison of the 427 reference (the GenBank database samples, Appendix A) mtDNA D-loop sequences and the Turkish native donkey populations sequences of 344 bp revealed 88 polymorphic sites (Appendix A). Considering all the variations, 145 haplotypes were derived, differing from each other by one to 24 variations. Twenty-four haplotypes were newly discovered in Turkish native donkeys (Hap 77–89, Hap 92, Hap 93, and Hap 95–103). Twenty-five haplotypes were shared among the Turkish native donkeys and those we collected from different countries’ donkeys (Hap 1–3, Hap 5, Hap 11, Hap 12, Hap 14, Hap 15, Hap 17–19, Hap 21, Hap 22, Hap 37, Hap 44, Hap 54, Hap 55, Hap 57, Hap 64–66, Hap 68, Hap 86, and Hap 90, Hap 91). While a common 25 haplotypes were known from Turkish native donkey populations and other countries’ donkeys, we identified two novel haplotypes in MRM region donkeys (CAT: Hap 81, Hap 82), four in BSR region donkeys (MER: Hap 83–85; KAS: Hap 87), two in CAR region donkeys (KON: Hap 88, Hap 89), two in MDR region donkeys (ANT: Hap 92, Hap 93), one in EAR region donkeys (KAR: Hap 95), and eight in SAR region donkeys (SAR: Hap 96–103) (Appendix A). One haplotype belonged to the MRM and BSR region donkeys (Hap 77: KIR, MER), one to MRM and MDR region donkeys (Hap 78: KIR, ANT), one to MRM and AER region donkeys (Hap 79: CAT, MUG), and one to MRM, MDR, and EAR region donkeys (Hap 80: CAT, ANT, KAR). The haplotype and nucleotide diversity of the mtDNA D-loop in different countries’ donkeys (427 reference mtDNA D-loop sequences) and Turkish indigenous donkeys are shown in Table 5.

The phylogenetic relationship among the 145 identified haplotypes was analyzed through the median-joining network. The MJN showed that there are five distinct lineages, as indicated in the phylogenetic tree and the star-like phylogeny (Figure 7a). The haplotypes of each population are detected by the color code, the abundance by the relative size of the symbol, and the diffusion among the breeds with the pie division of the different colors (Figure 7b,c).

The MJN analysis detected 145 haplotypes from 742 sequences (427 deposited GenBank database sequences (Appendix A) and 315 Turkish donkeys) showing a high variability (H_D_ = 0.890 ± 0.009). The median joining network showed that there are five distinct lineages, as shown in the phylogenetic tree and the star-like phylogeny. Three samples, Hap 126–127 and Hap 63, were horses (*E. cabullus*) that grouped together as an outgroup to the donkeys in the network. *Equus cabullus*, *Equus kiang,* and Asiatic wild asses (*E. hemionus luteus, E. hemionus kulan, E. Hemionus onager*) shared the (I) and (II) lineage, and had their respective lineages. Lineage (I) (*E. cabullus*) has three haplotypes, lineage (II) (*E. hemionus luteus, E. hemionus kulan, E. Hemionus onager, E. kiang)* has eight haplotypes. Four haplotypes were determined in *E. asinus somalicus,* and were grouped in the same (III) lineage. Lineages IV and V consist of domestic donkey populations. In Figure 6, Lineages IV and V, which have two different clades, are obviously defined in domestic donkey populations: clade 1 contains 50.68% of sequences, whereas clade 2 contains 49.31% of sequences in domestic donkey populations. Table 6 shows that genetic distances and nucleotide diversity of the two clades.

The AMOVA analysis is a useful tool to check how maternal genetic diversity is distributed within, and among, populations whose structure is quantified by F_ST_ values. In this study, we checked possible structures by creating and comparing different groups of populations. The analysis was made with two hypotheses: Hypothesis 1; the geographical distribution of the countries was considered. The domestic donkey populations were clustered in five groups: (1) Turkey’s indigenous donkey populations (n = 315), (2) European countries data: Balkan donkeys [26], Italian donkeys [28], Serbian donkeys [15] (n = 112), (3) Asian countries data: China, Tadzhikistan, Kyrgyzstan, Iran, Mongolia donkeys (n = 219) [17,30], (4) African origin domestic donkeys, and Kenyan donkeys, (n = 85) [16], (5) wild asses and others (n = 11) [11,36,37,38,39,40,41]. In hypothesis 2, the F_ST_ values and genetic distances (Reynolds’ linearized genetic distances in Turkish donkey populations from different regions) were considered between Turkey’s domestic donkey populations. (1) CAR, AER, and SAR region donkeys; (2) MDR, MRM, and EAR region donkeys, and (3) BSR region donkeys. Table 7 shows the results of the AMOVA analysis according to these hypotheses.

Hypothesis 1; the AMOVA analyses results revealed that the variation among groups, among populations within groups, and within populations were 7.95%, 1.72%, and 90.34%, respectively. The findings showed that most of the obtained variance is due to differences within populations. Variance components among groups were nonsignificant, demonstrating non-significant geographical distribution in the analyzed donkey populations. Moreover, the variance component among populations, within groups (*p* < 0.05), and within populations (*p* < 0.001) were significant.

Hypothesis 2; the AMOVA analysis findings indicated that the differentiation among groups, among populations within groups, and within populations were 7.31%, 0.39%, and 92.29%, respectively. The findings revealed that most of the obtained variance is based on differences within populations. Variance components among groups were significant (*p* < 0.05), which showed important geographical distribution in the analyzed Turkish donkey populations by using Reynolds’ linearized genetic distances. In addition, variance components among populations within groups were non-significant statistically, but were statistically significant within populations (*p* < 0.001).

## 4. Discussion

As exposed in some of genetic diversity studies in donkeys [13,19,24,25,30,42,43], the D-loop region of maternally ancestral mtDNA supplies adequate evidence to assess population genetic variation, evolutionary relationships, and matrilineal genetic origin of a species under consideration. This study presents the substantial analysis of mtDNA diversity in Turkish indigenous donkeys and supplies knowledge about the maternal lineage origins and genetic diversity of donkey populations, and so understanding into their genetic history.

### 4.1. Phylogenetic Relationships of Turkish Indigenous Donkey Populations

In the present study, mtDNA D-loop region from 315 Turkish indigenous donkeys were identified to clarify their phylogenetic relationship and haplogroups, and to define mtDNA haplotypes and their maternal ancestry. The results showed quite high genetic diversity within and between 16 Turkish donkey populations from the seven different geographical regions. By sequence alignment, 54 distinct haplotypes were defined among Turkish donkey populations, 28 of which are unique and 26 are shared among different donkey populations. No congruence of haplogroup to a population’s geographic ancestry was observed. Similarly to other research, the content of A+T was the richest in the mtDNA D-loop in Turkish donkey populations [28,44]. The haplotype distribution by clusters, corresponding to the Turkish native donkeys, showed two different clusters by using the maximum likelihood method (Figure 2b). These results were similar to Çınar Kul et al. [27] and Yalçın’s [45] studies.

Genetic diversity is commonly measured by haplotype (HD) and nucleotide diversity (πD), both of which reflect the magnitude of genetic variation at different scales [46,47]. The comparative result of H_D_ was consistent with that of π_D_. We observed moderate-to-high levels of haplotype diversity in Turkish donkey populations (ranging from 0.533 ± 0.180 to 0.933 ± 0.122) and moderate nucleotide diversity (ranging from 0.01196 ± 0.0026 to 0.02101 ± 0.0041) indicating plentiful genetic diversity in all the Turkish donkey populations. A high level of genetic variability was observed in the AYD populations followed by KAS, MUG, ISP, KON, and KUT. The overall values for haplotype and nucleotide diversity among all donkey populations was (HD: 0.763 ± 0.023, πD: 0.01680 ± 0.00046, respectively) similar to that estimated in other donkeys, such as the Italian donkey population donkeys (HD:0.862, πD: 0.018) [28], Anatolian donkey populations (HD: 0.756 ±0.0500, π_D_: 0.1688 ± 0.0012), Cypriot donkey populations (H_D_: 0.524 ± 0.209, π_D_: 0.00176 ± 0.001) [27], Andaluza donkey populations (H_D_: 0.737; π_D_: 0.0169) [4], and Somali wild asses (H_D_: 0.7417 + 0.0444) [18].

When haplotype and nucleotide diversity values for D-loop were compared with different donkey breeds, Turkish donkey populations were found to be lesser than, Ethiopian donkeys (H_D_: 0.903 ± 0.032; π_D_: 0.020 ± 0.003), reported by Kefena et al. [19], Balkan donkeys (H_D_: 0.982 ± 0.002; π_D_: 0.017 ± 0.009) reported by Pérez-Pardal et al. [26], Chinese donkeys (H_D_: 0.9055 ± 0.017–0.9778 ± 0.0540, π_D_: 0.02265 ± 0.00040–0.0285 ± 0.0160) reported by Chen et al. [17], Gan et al. [25], and Lei et al. [24]; Northeast African and South American donkeys (H_D:_ 0.737 ± 0.028–0.910 ± 0.032; π_D_: 0.0058 ± 0.0008–0.0179 ± 0.0035) reported by Xia et al. [42], and the Balkan donkey from Serbia (H_D_: 0.849 ± 0.087; π_D_: 0.01549 ± 0.008) reported by Stanisic et al. [15], while they were found to be higher than Spanish donkeys (H_D_: 0.421, π_D_: 0.0006) reported by Aranguren-Mendez et al. [4].

Pairwise F_ST_ values between Turkish native donkey populations and geographical regions were estimated. Most of the populations show little differentiation, with F_ST_ < 0.05 (*p* > 0.01) or moderate, but no significant differentiation (F_ST_ > 0.05, *p* > 0.01). However, great differentiation was found between AER, MRM, MDR, BSR, and CAR region populations (KUT-KIR, KUT-CAT, KUT-KRM, KRM-AYD, KRM-KAS, KAS-CAT, KON-CAT, and KON -KRM, all F_ST_ values (F_ST_ > 0.25) (*p* < 0.01). Matrix of pairwise F_ST_, Reynolds linearized pair-wise matrilineal genetic distances and D_A_ genetic distances confirmed similar results between 16 Turkish native donkey populations. A Neighbor joining tree (NJT) from Reynolds’ linearized genetic distances clearly demonstrated three distinct clades in Turkish donkey populations from seven different regions; Clade I (CAR, AER, SAR); Clade II (MDR, MRM, and EAR), Clade III (BSR). NJT reconstructed using D-loop sequence F_ST_ and D_A_ genetic distance value estimates also generated the same topology (figure not shown) to the dendrogram tree constructed using Reynolds’ linearized pair-wise genetic distance, shown in Figure 3. The pairwise F_ST_ values of Turkish donkey populations detected in this study were similar to the value of Mexican Creole donkeys [13], and Spanish donkey breeds [4], but lower than of the Brazilian donkey population, Peruvian donkey population, Ethiopian donkey population [42], Italian donkey populations [28], and Chinese donkeys [25].

The MJN analysis retrieved two major clades among the 16 different Turkish donkey populations analyzed in the present study. Two clades, separated from five nucleotide substitutions each, were found. These clades comprise haplotypes for which common origins are assumed since they share a characteristic pattern of mutations. As revealed in the network, it clearly showed two lineages and indicated a star like phylogenetic pattern, in which two large haplotypes, Hap 1 and Hap 4, are in the center of clade 1 and clade 2 lineages, respectively. The clade 2 lineage predominated slightly (60.635%, 191/315). D-loop gene lineages in Turkish donkeys appear mixed with those of other donkeys.

### 4.2. Phylogenetic Relationships between Turkish Native Donkeys and Other Donkeys

The Turkish native donkey mtDNA sequences (n = 315) were compared with 427 publicly available mtDNA D-loop sequences belonging to different countries’ donkey breeds, wild asses [11,15,16,17,23,29,30,31,32,33,34,35], and the horses (*Equus cabullus*, *Equus kiang)* as outgroups [16,30]. The aim of the comparisons with public mtDNA D-loop sequences was to identify the closely related populations among the Turkish native donkeys and the other worldwide donkey breeds.

The haplotype and nucleotide diversity within Turkish native donkeys were (0.763 ± 0.023) and (0.01680 ± 0.00046), respectively. The haplotype and nucleotide diversity values of other countries’ donkeys ranged from (0.890 ± 0.013), (0.01759 ± 0.00053) in the Chinese donkeys to (0.983 ± 0.007), and (0.02843 ± 0.00195) in the African origin domestic donkeys, respectively. The haplotype and nucleotide diversity values of the Turkish donkey populations was found to be lower than that of the other donkey populations (Table 5; Balkan, Chinese, Serbian, Italian etc.), indicating a relatively low diversity. This finding is consistent with other genetic diversity studies in different countries’ donkey populations [15,26,28]. The haplotype diversity found here was higher than that found in a previous study in three different Turkish donkey populations [27], and the nucleotide diversity found here was lower than that found in that previous study. These results indicate a relatively higher level of genetic diversity in the 16 Turkish donkey populations compared with other countries’ donkey populations. For example, the haplotype diversity and nucleotide diversity values of 10 Balkan donkey populations were 0.982 ± 0.002 and 0.017 ± 0.009 [26]. However, according to Walsh’s work on the required sample size for the diagnosis of conservation units a sample of 59 individuals is necessary to reject the hypothesis that individuals with unstamped (“hidden”) character states exist in the population size. Thus, the sample size necessary to reject a hidden state frequency of 0.05 is 56, when sampling from a finite population of 500 individuals [48]. Our genetic diversity estimation is therefore a precise reflection of Turkish indigenous donkeys due to the large sample size used in this study.

The MJN analysis identified 145 haplotypes from 742 mtDNA D-loops (427 Genbank sequences and our 315 Turkish donkeys) based on 344 bp of the control region sequence, and all these haplotypes grouped into five lineages. *Equus cabullus*, *Equus kiang,* and Asiatic wild asses (*E. hemionus luteus, E. hemionus kulan, E. Hemionus onager*) shared (I) and (II) lineages and had their respective lineages. Lineage (I) (*Equus cabullus*) have three haplotypes, lineage (II) (*E. hemionus luteus, E. hemionus kulan, E. Hemionus onager, Equus kiang)* has eight haplotypes. Four haplotypes were identified in *E. asinus somalicus* and were distributed into the same lineage (III). There were 130 haplotypes in domestic donkey populations representing two lineages (IV-clade I, V-clade II). Out of 145 haplotypes, 54 (clade I) are referred to Somali lineages (*Equus asinus somalicus*), whereas 75 haplotypes (clade II) belong to Nubian lineage (*Equus africanus africanus*). The clade I haplotypes were mainly derived from a single major haplotype (Hap 2), with a simple star-like shape, whereas the genetic architecture of the clade II haplotypes was more complicated, with more universally occurring haplotypes (e.g., Hap 5, Hap 37, Hap 65). This was consistent with the much higher genetic distance and nucleotide diversity within the clade II lineage than within the clade I lineage, implying that the clade II lineage involved many more individuals at the beginning of domestication compared with the clade I lineage. In the study twenty-four haplotypes were newly discovered in Turkish native donkeys (Hap 77–89, Hap 92, Hap 93, and Hap 95–103). When we compared all sequences, twenty-five haplotypes were shared among the different countries’ donkey breeds and Turkish native donkey populations (Hap 1–3, Hap 5, Hap 11, Hap 12, Hap 14, Hap 15, Hap 17–19, Hap 21,Hap 22, Hap 37, Hap 44, Hap 54, Hap 55, Hap 57, Hap 64–66, Hap 68, Hap 86, Hap 90, Hap 91) (11,15–17,26,28,30,37,38). The MJ network analysis showed that both Somali and Nubian lineages had made a genetic contribution to Turkish indigenous donkey evolution, as 19 haplotypes of lineage Somali (n = 191 samples) and 30 haplotypes of lineage Nubian (n = 124) were identified in Turkish donkey populations. These findings showed that 16 different Turkish native donkey populations and other domestic donkey populations possessed abundant mtDNA diversity, and indicated two different maternal origins. Clade 1 includes 49.59% of sequences, whereas clade 2 includes 48.52% of sequences. Out of 145 haplotypes, 53 were referred to the Somali lineage (clade I), whereas 77 haplotypes belonged to Nubian lineage (clade II). As shown in the network, two lineages were clearly revealed, and showed a star like phylogenetic pattern, in which two large haplotypes, H2 and H5, are located in the center of the clade 1 and clade 2 lineages, respectively. The clade 1 lineage appeared to predominate slightly (49.59%, 368/742) in the domestic donkeys. The results showed that the Nubian (*E. africanus africanus*) and Somali (*E. asinus somali*) wild asses were the probable progenitor for ancient Turkish donkeys and domestic donkeys. These results support previous studies on the origins of the domestic donkey [15,16,17,26,28,37].

In our study, to assess a hierarchical structure among Turkish native donkey populations and other countries’ donkeys (Italian, Serbian, Chinese, etc.) and wild asses, AMOVA analysis was performed under two different hypotheses, grouping populations in different clusters. In both hypotheses, the AMOVA analysis results indicated that the majority of the observed variance was due to differences among individuals within populations. The greater part of the variation was observed within individuals (90.34% hypothesis 1 and 92.29% hypothesis 2) whereas the differences among groups represented only the 7.95% (hypothesis 1) and 7.31% (hypothesis 2) of the variation, respectively. Our results are similar to other studies on the mtDNA D-loop region [15,26,28,42]. In hypothesis 1; among populations within groups and among individuals within populations there was statistically significant variation (*p* < 0.001, *p* < 0.05). In hypothesis 2; among groups and among individuals within populations there was also statistically significant variation (*p* < 0.001, *p* < 0.05). Our AMOVA analysis and phylogenetic analysis revealed low genetic variation between the groups that were defined geographically among Turkish donkey populations. It was observed that there is a small difference in the donkey population of the BSR compared to the other region’s populations. In order to explain the reason for this difference, it is considered appropriate to conduct new studies with more samples in BSR.

## 5. Conclusions

The present study demonstrated the abundant mtDNA diversity existing in Turkish indigenous donkey populations. The detection of 54 haplotypes in 315 donkeys suggests that abundant genetic variety exists in Turkish donkey populations. We confirmed two different maternal lineages (Somali and Nubian) of domestic donkeys reported by other researchers. No obvious geographical distribution was found among Turkish donkey populations. The study found high nucleotide and haplotype diversity values, no haplotypes clearly distinct from other populations, and no clear clustering on the median joining tree in Turkish donkeys. In summary, our results imply the moderately high genetic diversity of the 16 Turkish donkey populations in mtDNA D-loop region, and give an insight into the origin of the analyzed populations. Our results obviously exclude the Asian wild donkey as ancestors of Turkish indigenous donkey populations, and the two African wild donkeys (Somali and Nubian) are the likely ancestors of Turkish indigenous donkeys. The results provided here could be regarded as a genetic structure of the maternal ancestors of Turkish donkey populations.

## Figures and Tables

**Figure 1 animals-10-01970-f001:**
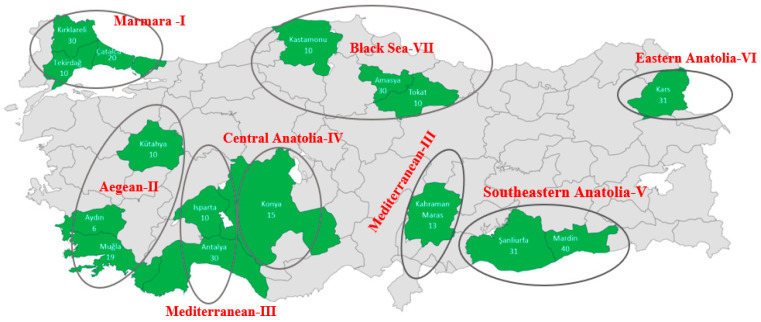
Turkish indigenous donkey populations, sampling areas from the seven different geographical regions of Turkey.

**Figure 2 animals-10-01970-f002:**
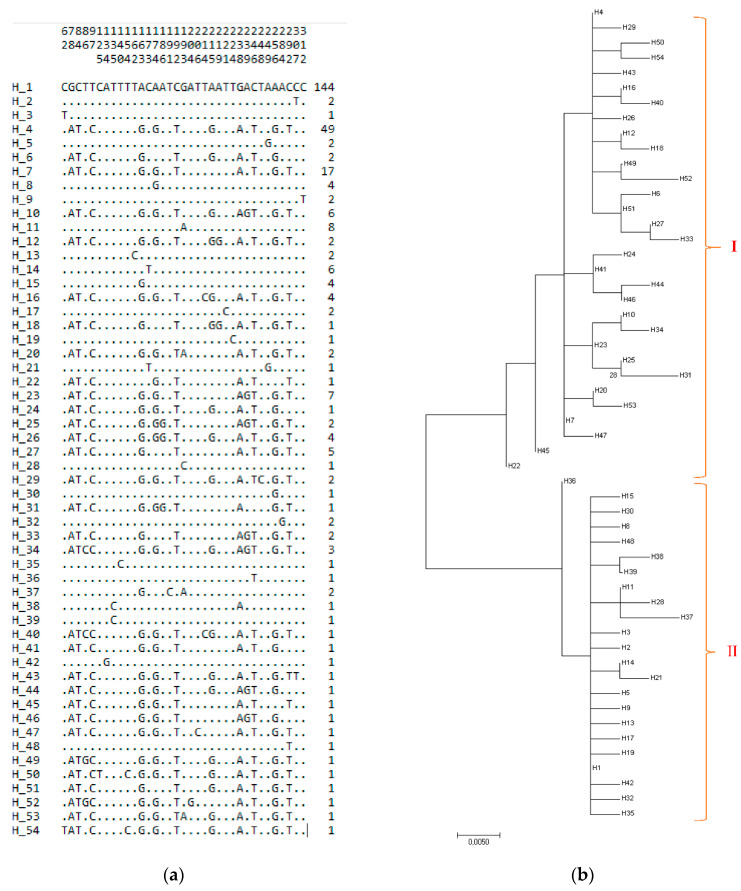
(**a**) Polymorphic sites and haplotypes, (**b**) neighbor joining tree by maximum likelihood method distribution of haplotypes in Turkish indigenous donkeys.

**Figure 3 animals-10-01970-f003:**
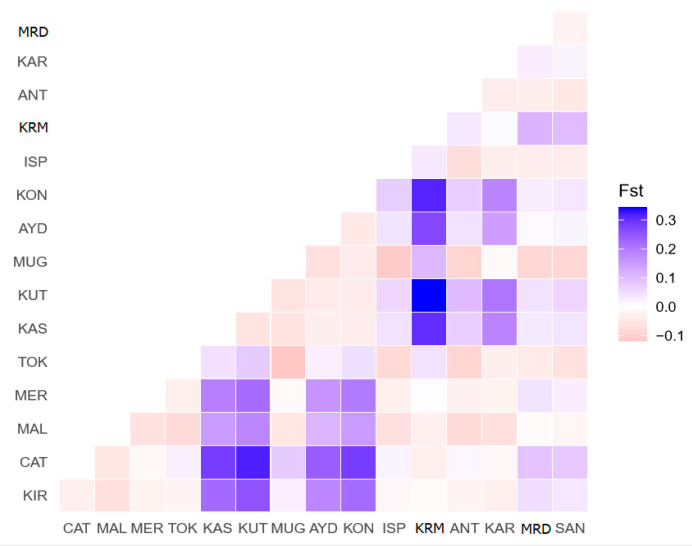
Pairwise F_ST_ values for the 16 Turkish indigenous donkey population comparisons.

**Figure 4 animals-10-01970-f004:**
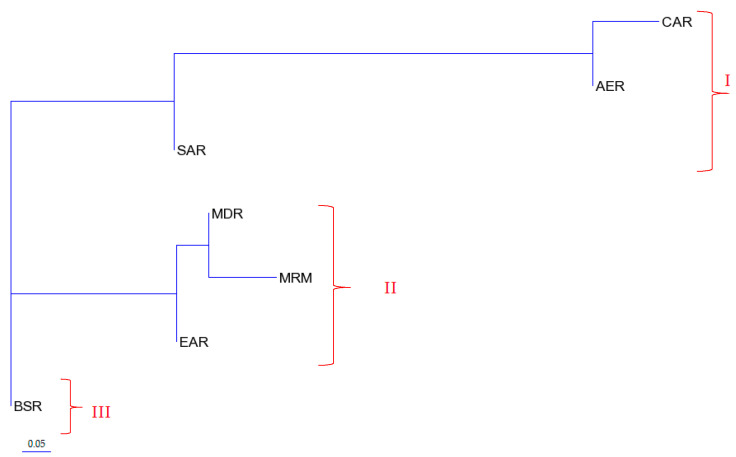
Neighbor-joining tree from Reynolds’ linearized genetic distances in Turkish donkeys from different regions. (CA: Central Anatolia; AEG: Aegean; SEA: Southeastern Anatolia; MED: Mediterranean; MAR: Marmara; EA: Eastern Anatolia, BS: Black Sea).

**Figure 5 animals-10-01970-f005:**
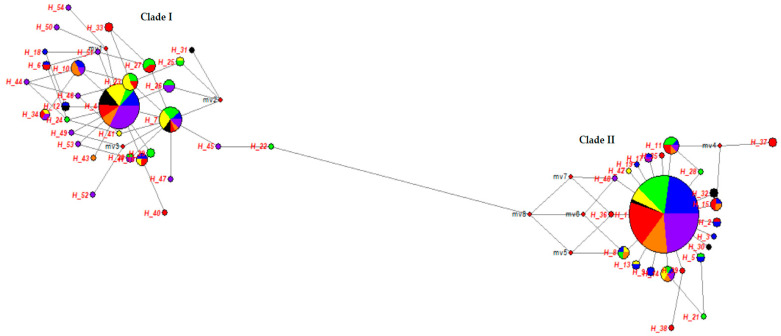
Median-joining network constructed from 54 haplotypes obtained from seven different geographical regions and 16 different populations (MRM: Blue, BSR: Green, AER: Yellow, MDR: Red, CAR: Black, EAR: Orange, SAR: Purple).

**Figure 6 animals-10-01970-f006:**
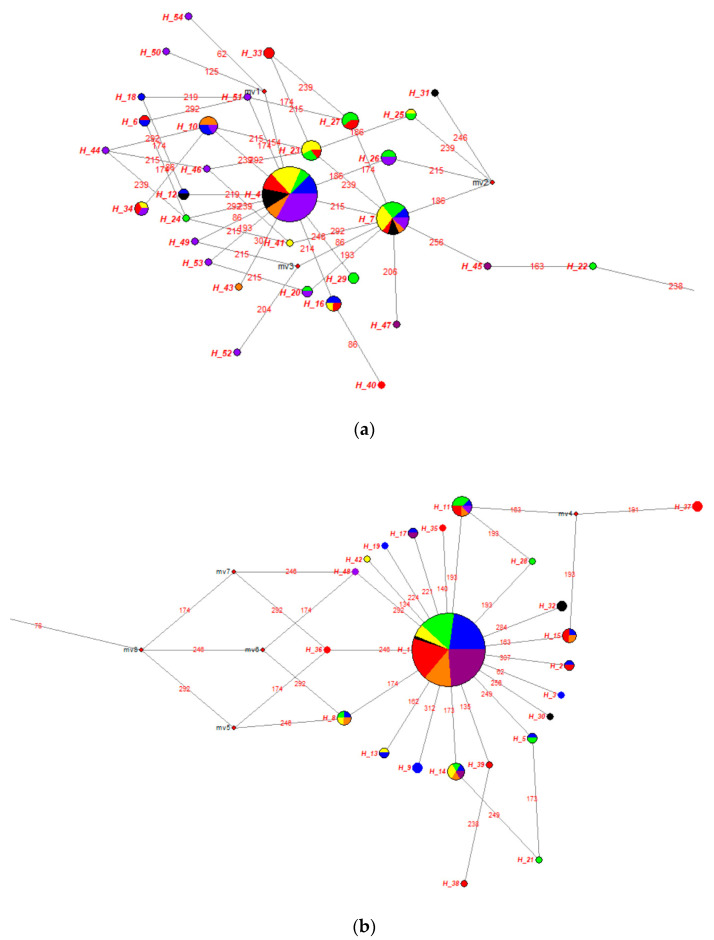
Clade I (**a**) and Clade II (**b**) median-joining network constructed from 54 haplotypes obtained from seven different geographical regions and 16 different populations (MRM: Blue, BSR: Green, AER: Yellow, MDR: Red, CAR: Black, EAR: Orange, SAR: Purple).

**Figure 7 animals-10-01970-f007:**
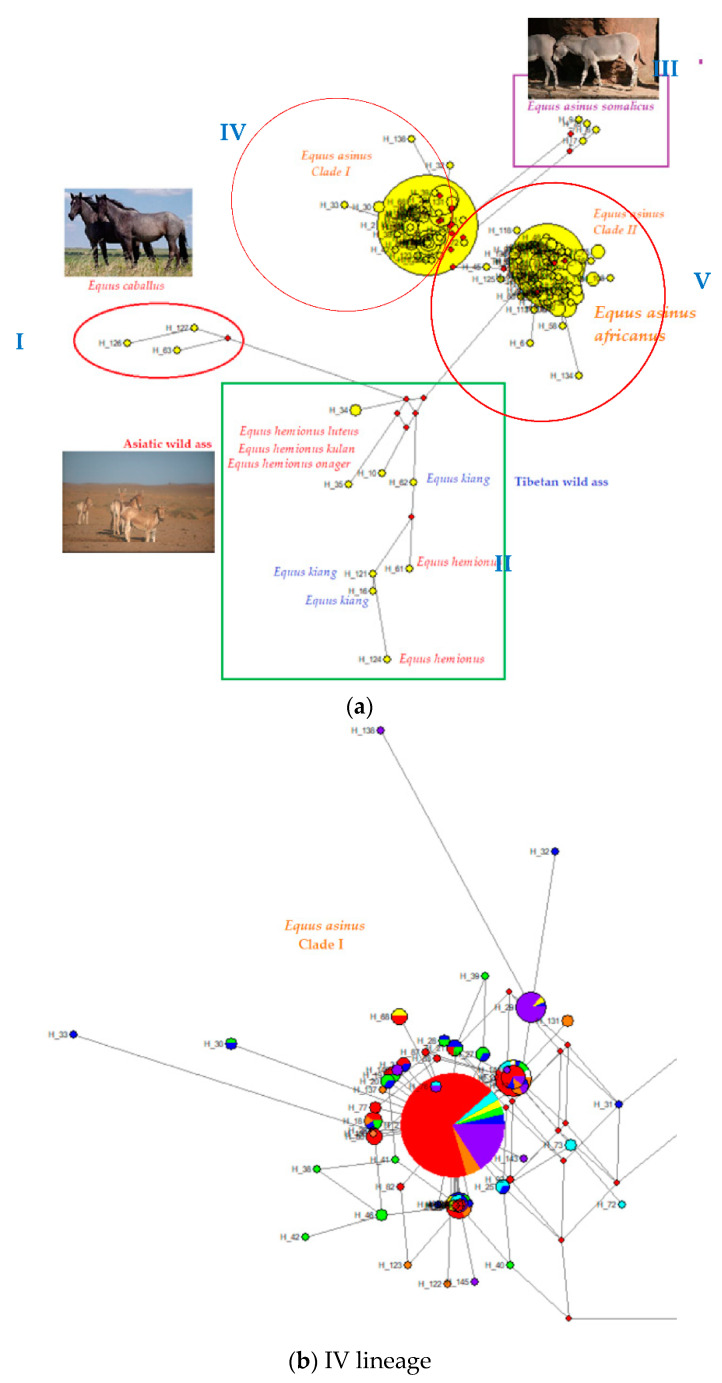
(**a**) MJN based on the 16 different regions of Turkish native donkey populations and the 429 deposited sequences of donkey from GenBank database. Circles are proportional to the haplotypes frequencies; black circles are median vectors (mv), performing extant unsampled or extinct ancestral sequences; in red the number of mutation points with respect to the reference sequence X97337. (**b**) IV; Clade I: Somali lineages (*Equus asinus somalicus*), (**c**) V; Clade II: Nubian lineage (*Equus africanus africanus*). (Red: Turkish native donkey populations; Blue: [16], Green: [26], Yellow: [15], Orange: [30], Purple: [17], White: [11], Turquoise: [28], Grey: [38]).

**Table 1 animals-10-01970-t001:** The sampled regions, provinces, and the geographical coordinates, with the number of individuals.

Sampled Regions	Provinces	Geographical Coordinate	Number of Individuals
(I)-Marmara (MRM)	KIR	41°51′ N 27°19′ E	30
	CAT	41°06′ N 28°30′ E	20
	MAL	40°52′ N 26°57′ E	10
(VII)-Black Sea (BSR)	MER	40°53′ N 35°32′ E	30
	TOK	40°12′ N 36°27′ E	10
	KAS	41°50′ N 32°54′ E	10
(II)-Aegean (AER)	KUT	39°21′ N 30°01′ E	10
	MUG	36°37′ N 29°26′ E	19
	AYD	37°44′ N 28°01′ E	6
(IV)-Central Anatolia (CAR)	KON	37°38′ N 32°26′ E	15
(III)-Mediterranean (MDR)	ISP	37°49′ N 30°44′ E	10
	KRM	37°30′ N 36°57′ E	13
	ANT	36°50′ N 30°13′ E	30
(VI)-Eastern Anatolia (EAR)	KAR	40°36′ N 43°07′ E	31
(V)-South East Anatolia (SAR)	MRD	37°18′ N 40°44′ E	40
	SAN	37°10′ N 38°50′ E	31
		Total	315

KIR: Kırklareli, CAT: İstanbul-Çatalca, MAL: Tekirdağ-Malkara, MER: Amasya-Merzifon; TOK: Tokat, KAS: Kastamonu-Cide, KUT: Kütahya, MUG: Muğla, AYD: Aydın, KON: Konya, ISP: Isparta, KRM: Kahramanmaraş, ANT: Antalya, KAR: Kars, MRD: Mardin, SAN: Şanlıurfa.

**Table 2 animals-10-01970-t002:** Genetic characterization of the 16 Turkish donkey populations.

Provinces	n	N_H_	H_D_ ± SD	S	SPI	S_S_	P_H_	S_H_	π_D_ ± SD	k	Region
**KIR**	30	12	0.710 ± 0.007	18	11	7	1	11	0.01393 ± 0.0025	4.848	MRM
**CAT**	20	11	0.763 ± 0.0106	19	11	8	2	9	0.01288 ± 0.0036	4.495	
**MAL**	10	4	0.533 ± 0.180	12	11	1	-	4	0.01496 ± 0.0044	5.222	
**MER**	30	13	0.720 ± 0.091	16	15	1	3	10	0.01446 ± 0.0023	5.048	BSR
**TOK**	10	5	0.756 ± 0.130	13	12	1	-	5	0.01821 ± 0.0032	6.356	
**KAS**	10	7	0.911 ± 0.077	14	12	2	2	5	0.01731 ± 0.0036	6.022	
**KUT**	10	4	0.778 ± 0.091	13	12	1	-	4	0.01681 ± 0.0038	5.867	AER
**MUG**	19	9	0.871 ± 0.048	16	11	5	2	7	0.01696 ± 0.0020	5.918	
**AYD**	6	5	0.933 ± 0.122	14	10	4	-	5	0.02101 ± 0.0041	7.333	
**KRM**	13	4	0.654 ± 0.106	12	11	1	-	4	0.01646 ± 0.0026	5.744	MDR
**ISP**	10	7	0.867 ± 0.107	15	12	3	1	6	0.01917 ± 0.0033	6.689	
**ANT**	30	12	0.717 ± 0.090	18	15	3	5	7	0.01196 ± 0.0026	4.172	
**KON**	15	7	0.829 ± 0.082	14	12	2	3	4	0.01659 ± 0.0027	5.791	CAR
**KAR**	31	9	0.652 ± 0.091	15	12	3	1	8	0.01455 ± 0.0023	5.079	EAR
**MRD**	40	15	0.758 ± 0.060	18	12	6	8	7	0.01753 ± 0.0008	6.118	SAR
**SAN**	31	9	0.695 ± 0.076	17	13	4	3	6	0.01764 ± 0.0012	6.155	
	315	54	0.763 ± 0.023	35	29	6			0.01680 ± 0.00046	5.745	

Sample size (n), number of haplotypes (N_H_), haplotype diversity (H_D_), with their standard deviation (SD), total polymorphic sites (S), parsimony informative sites (SPI), singleton site (S_S_), private haplotype (P_H_), shared haplotype (S_H_), nucleotide diversity (π_D_) with their SD and average number of nucleotide differences (k) within and across the 16 populations.

**Table 3 animals-10-01970-t003:** The number of haplotypes observed by regions.

Region	N_H_	Haplotypes
**MRM**	19	Hap 1-19
**BSR**	17	Hap 1, Hap 4, Hap 5, Hap 7, Hap 8 Hap 11, Hap 14, Hap 20 -29
**AER**	12	Hap 1, Hap 4, Hap 7, Hap 8, Hap 13, Hap 14, Hap 16, Hap 23, Hap 25, Hap 34, Hap 41, Hap 42
**MDR**	18	Hap 1, Hap 2, Hap 4, Hap 6, Hap 7, Hap 11, Hap 15, Hap 16, Hap 23, Hap 27, Hap 33-40
**CAR**	7	Hap 1, Hap 4, Hap 7, Hap 12, Hap 30-32
**EAR**	9	Hap 1, Hap 4, Hap 7, Hap 8, Hap 10, Hap 11, Hap 14, Hap 15, Hap 43
**SAR**	21	Hap 1, Hap 4, Hap 7, Hap 10, Hap 11, Hap 14, Hap 17, Hap 20, Hap 26, Hap 34, Hap 44-54.

The number of haplotypes (N_H_).

**Table 4 animals-10-01970-t004:** Matrix of pairwise F_ST_ (below diagonal) and D_A_ (D-loop nucleotides sequence divergence, above diagonal) distances between 16 Turkish indigenous donkey populations.

Sampling Regions	MRM	BSR	AER	MDR	CAR	EAR	SAR
**MRM**	-	0.00036	0.00391	−0.00014	0.00465	−0.00024	0.00115
**BSR**	0.0242	-	0.00139	0.00011	0.00190	−0.00007	0.00001
**AER**	0.2106 **	0.0765 *	-	0.00323	−0.00049	0.00273	0.00052
**MDR**	−0.0099	0.0068	0.1734 **	-	0.00399	−0.00026	0.00082
**CAR**	0.2523 **	0.1027 *	−0.0302	0.2130 **	-	0.00340	0.00069
**EAR**	−0.0171	−0.0052	0.01464 *	−0.0184	0.1843 **	-	0.00047
**SAR**	0.0685 *	0.0004	0.0291	0.04799 *	0.0376	0.0263	-

* *p* < 0.05; ** *p* <0.01.

**Table 5 animals-10-01970-t005:** The haplotype and nucleotide diversity of mtDNA D-loop within different countries’ donkey populations.

Population	n	H_D_ ± SD	π_D_ ± SD	References
Turkish native donkeys	315	0.763 ± 0.023	0.01680± 0.00046	This study
Different countries’ donkeys	10	0.956 ± 0.059	0.02349 ± 0.00598	[11,36,37,38,39,40]
African origin domestic donkeys	85	0.983 ± 0.007	0.02843 ± 0.00195	[16]
Chinese donkeys	146	0.890 ± 0.013	0.01759± 0.00053	[17]
Balkan donkeys	62	0.973 ± 0.011	0.01764 ± 0.00067	[26]
Serbian donkeys	23	0.949 ± 0.033	0.01922 ± 0.00136	[15]
Nigerian, Iranian, Tadzhikistan, Chinese, Kenya, and Kyrgyzstan donkeys	73	0.965 ± 0.010	0.02613 ± 0.00374	[30]
Italian donkeys	28	0.913 ± 0.033	0.01568 ± 0.00133	[28]

**Table 6 animals-10-01970-t006:** Genetic distance and nucleotide diversity of the two clades.

Population	Pairwise Distance ± SD	π_D_ ± SD
Clade I lineage	1.4228 ± 0.8715	0.004136 ± 0.002802
Clade II lineage	2.1506 ± 1.1992	0.00525 ± 0.003857

**Table 7 animals-10-01970-t007:** The AMOVA analysis among the Turkish native donkey populations and 427 reference mtDNA D-loop sequences.

Source of Variation	Variance Component	Variance (%)	Fixation Index ^a^	*p*-Value ^b^
Hypothesis 1: 5 different clusters				
Among groups	0.31406	7.95	**Φ**_CT_: 0.07945	0.08211 ^ns^
Among populations within groups	0.06789	1.72	**Φ**_SC_: 0.01866	0.0303 *
Within populations	3.57081	90.34	**Φ**_ST_: 0.09663	0.000 ***
Hypothesis 2: Turkish native donkey populations; genetic distances tree				
Among groups	0.21621	7.31	**Φ**_CT_: 0.07314	0.0303 *
Among populations within groups	0.01157	0.39	**Φ**_SC_: 0.00422	0.3304 ^ns^
Within populations	2.72850	92.29	**Φ**_ST_: 0.07705	0.000 ***

^a^**Φ**_CT_: variation among groups divided by total variation; **Φ**_SC_: variation among sub-groups divided by the sum of variation among sub-groups within groups and variation within sub-groups; **Φ**_ST_: the sum of variation groups divided by total variation. ^b^ ns = *p* > 0.05; * *p* < 0.05; *** *p*< 0.001.

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
