# Peer review of "Phylogenetic Relationships of Turkish Indigenous Donkey Populations Determined by Mitochondrial DNA D-loop Region"

_animals, 2020, doi:10.3390/ani10111970_

Round 1

Reviewer 1 Report

The authors of the manuscript entitled “Phylogenetic relationships of Turkish Indigenous Donkey Populations determined by mitochondrial DNA D-loop region” performed a phylogenetic analysis of indigenous Donkey populations and compared the haplotype diversity among these populations and among populations from other regions. The results presented are interesting and the methodology scientifically sounds. However, prior to publication some improvement must be performed regarding the writing and the structure of the manuscript. The English must be reviewed. There are several moments where the text is confusing, or the not most accurate grammar is applied. However, my main concern is about the length of the manuscript. The manuscript is very long and must be summarized. For example, the introduction is very long. The authors should reduce it substantially removing some information which are not relevant for the moment. For example, lines 49-75, the authors could summarize all the discussion about the D-loop structure and the comparison with the horse sequence. This information is well established in the literature and don’t need to be rewrite at this moment. The results can also be summarized. For example, there are complete sentences repeated between the results and discussion sections, such as Lines 419-422.

Additionally, the authors should provide the number of sequences for each Donkey population obtained from NCBI (Lines 167-170) even with similar information present on Table S1. On lines 174-176 the authors inform the use of two approaches (neighbor joining tree reconstruction and the median-joining network (MJN)). However, three software are described as used for these analyses. The authors cannot use the word “respectively” at the end of this sentence. Please, provide the correct link between each approach and the software used for the analysis.

“In this study, it is concluded that mtDNA D-loop region sequence analysis is the most informative tool for the identification of genetic variety, origin, and phylogeny of different donkey populations”. The authors cannot conclude this. There is not data in the literature to compare the efficiency of other methods to describe the genetic diversity in Donkeys. Actually, using other species as model, there are evidence of high efficiency of high-throughput methods for identification of genetic diversity.

Minor comments:

Lines 23-26: The sentence is not clear. Avoid starting the sentence with “This research”.

Line 27: Insert a period (.) after “defiend” and start a new sentence.

Line 31: Remove “each with more than 10 samples”, the percentages are already informed.

Lines 52-53: The genetic origins of what?

Line 68: “One less” what? A nucleotide?

Line 85: Remove extra space.

Lines 103-105: Add the correspondent references.

Line 113: Add a comma before (iv).

Line 117: There is misplaced sentence at the end of the introduction.

Line 137: There is a misplaced sentence before the Table 1. “The isolated DNA was diluted with TE buffer.”

Line 184: remove extra space after “Hap 1”.

Line 186: The authors are calling the supplementary Table 1 with different names (Table S1 and Supplementary Materials Table 1). Please, keep the consistency.

Line 280-282: The authors begun the sentence with “As most of the…”. This structure suggests the introduction of an additional information at the end of the sentence. However, the sentence seems to finish before this information. For example, “As most of the the highest FST (> 0.25) were seen between XXX and XXX, we would expect (or any other information)”.

Lines 292-293: This sentence is not grammatically correct.

Figure 5: This figure has a very low resolution and there are misplaced sentences included in the figure (Clade I and Clade II in red).

Table 6: The lower boarder of the table is missing.

Author Response

Response to Reviewer-1 Comments:

The authors of the manuscript entitled “Phylogenetic relationships of Turkish Indigenous Donkey Populations determined by mitochondrial DNA D-loop region” performed a phylogenetic analysis of indigenous Donkey populations and compared the haplotype diversity among these populations and among populations from other regions. The results presented are interesting and the methodology scientifically sounds. However, prior to publication some improvement must be performed regarding the writing and the structure of the manuscript.

  • The English must be reviewed. There are several moments where the text is confusing, or the not most accurate grammar is applied: “revised as requested”
  • For example, lines 49-75, the authors could summarize all the discussion about the D-loop structure and the comparison with the horse sequence. This information is well established in the literature and don’t need to be rewrite at this moment: “revised as requested”
  • The results can also be summarized. For example, there are complete sentences repeated between the results and discussion sections, such as Lines 419-422: Necessary controls have been made and repeated parts have been removed: “revised as requested”
  • Additionally, the authors should provide the number of sequences for each Donkey population obtained from NCBI (Lines 167-170) even with similar information present on Table S1: “revised as requested”
  • On lines 174-176 the authors inform the use of two approaches (neighbor joining tree reconstruction and the median-joining network (MJN)). However, three software are described as used for these analyses. The authors cannot use the word “respectively” at the end of this sentence. Please, provide the correct link between each approach and the software used for the analysis: “revised as requested”
  • “In this study, it is concluded that mtDNA D-loop region sequence analysis is the most informative tool for the identification of genetic variety, origin, and phylogeny of different donkey populations”. The authors cannot conclude this. There is not data in the literature to compare the efficiency of other methods to describe the genetic diversity in donkeys. Actually, using other species as model, there are evidence of high efficiency of high-throughput methods for identification of genetic diversity: “revised as requested”

  • Minor comments: 

Lines 23-26: The sentence is not clear. Avoid starting the sentence with “This research”.: "This research" expression has been changed to "In this study". 

Line 27: Insert a period (.) after “defiend” and start a new sentence. Done

Line 31: Remove “each with more than 10 samples”, the percentages are already informed. Done.

Lines 52-53: The genetic origins of what? “Genetic origin” has been added.

Line 68: “One less” what? A nucleotide? “one less nucleotide” has been added.

Line 85: Remove extra space. It has been removed.

 Lines 103-105: Add the correspondent references. It has been added the references.

 Line 113: Add a comma before (iv). Done

Line 117: There is misplaced sentence at the end of the introduction. Done

Line 137: There is a misplaced sentence before the Table 1. “The isolated DNA was diluted with TE buffer.” It has been deleted.

Line 184: remove extra space after “Hap 1”. Done.

Line 186: The authors are calling the supplementary Table 1 with different names (Table S1 and Supplementary Materials Table 1). Please, keep the consistency. Done.

Line 280-282: The authors begun the sentence with “As most of the…”. This structure suggests the introduction of an additional information at the end of the sentence. However, the sentence seems to finish before this information. For example, “As most of the the highest FST (> 0.25) were seen between XXX and XXX, we would expect (or any other information)”. “revised as requested”

Lines 292-293: This sentence is not grammatically correct. Changed.

Figure 5: This figure has a very low resolution and there are misplaced sentences included in the figure (Clade I and Clade II in red). “revised as requested”

Table 6: The lower boarder of the table is missing. It has been added.

Reviewer 2 Report

Dear Authors, 

The manuscript under the title: "Phylogenetic relationships of Turkish Native Donkey Populations determined by mitochondrial DNA D-loop region" is very interesting and after some minor issues will be ready to publish in Animals.

Abstract
Needs to be shortened. For example ln 30-31 (...) each with .... please delete. Ln 37 Clade I and (...) and so on, please delete. ln 39-41 please rewrite to a shortened sentence. ln 44, replace the to this.

Introduction

ln 62. its-->mtDNA

ln 65 delete closely

ln 67: portion--> fragment

ln 67-68: sentence not clear please explain.

ln105-107: I feel that is not relevant.

material and methods

the sampled populations need to be far better described. 

ln 131:" judged in comparison replace to compared

ln 372: well, AMOVA is not a useful tool - in some cases is a gold standard. please rewrite

ln 382: will you please explain the idea of the hypothesis in the results? this should be in the introduction section

Author Response

Response to Reviewer’s-2 Comments

The manuscript under the title: "Phylogenetic relationships of Turkish Native Donkey Populations determined by mitochondrial DNA D-loop region" is very interesting and after some minor issues will be ready to publish in Animals.

Abstract

- Needs to be shortened. For example ln 30-31 (...) each with .... please delete. Ln 37 Clade I and (...) and so on, please delete. ln 39-41 please rewrite to a shortened sentence. ln 44, replace the to this. Done

Introduction

ln 62. its-->mtDNA: Done

ln 65 delete closely: Done

ln 67: portion--> fragment : “revised as requested”

ln 67-68: sentence not clear please explain. “revised as requested”

ln105-107: I feel that is not relevant. “revised as requested”

Material And Methods

the sampled populations need to be far better described.  Done

ln 131:" judged in comparison replace to compared: Done

ln 372: well, AMOVA is not a useful tool - in some cases is a gold standard. please rewrite “revised as requested”

ln 382: will you please explain the idea of the hypothesis in the results? this should be in the introduction section “revised as requested”
